# Spatiotemporal trends in bed bug metrics: New York City

**Kathryn P. Hacker** [ID]**[1,2]** *, **Andrew J. Greenlee[3]**, **Alison L. Hill[4]**, **Daniel Schneider[3]**, **Michael Z. Levy[1]**

**1** Department of Epidemiology, Biostatistics, and Informatics, University of Pennsylvania, Philadelphia, Pennsylvania, United States of America, **2** Department of Epidemiology, University of Michigan, Ann Arbor, Michigan, United States of America, **3** Department of Urban and Regional Planning, University of Illinois at Urbana-Champaign, Urbana, Illinois, United States of America, **4** Institute for Computational Medicine, Johns Hopkins University, Baltimore, Maryland, United States of America

* kphacker@umich.edu

**Data Availability Statement:** All data for this project is freely available from public governmental databases. Specifically the data from 2014-2019 can be downloaded directly from New York City Open Data including: Housing Maintenance Code

## Abstract

Bed bug outbreaks pose a major challenge in urban environments and cause significant strain on public resources. Few studies have systematically analyzed this insect epidemic or the potential effects of policies to combat bed bugs. Here we use three sources of administrative data to characterize the spatial-temporal trends of bed bug inquiries, complaints, and reports in New York City. Bed bug complaints have significantly decreased (p < 0.01) from 2014–2020, the absolute number of complaints per month dropping by half (875 average complaints per month to 440 average complaints per month); conversely, complaints for other insects including cockroaches and flies did not decrease over the same period. Despite the decrease of bed bug complaints, areas with reported high bed bug infestation tend to remain infested, highlighting the persistence of these pests. There are limitations to the datasets; still the evidence available suggests that interventions employed by New York City residents and lawmakers are stemming the bed bug epidemic and may serve as a model for other large cities.

## Introduction

Bed bugs (*Cimex lectularius*) have reemerged as a substantial public health and economic issue, particularly in dense urban environments [1–5]. While bed bugs were largely controlled after the Second World War, their populations have resurged since. The resurgence of bed bug populations is likely due to a combination of many factors, among these insecticide resistance [6–9], increased mobility, and exchange of used furniture [10–12]. By the 1990s, bed bugs were again documented globally as an arthropod pest of public health importance [1–3, 7, 13, 14].

The overall prevalence of bed bug infestation in major US cities is high, though rarely systematically measured, and has attracted the attention of the media and policy entities [14–16]. Trends in bed bug resurgence and control effectiveness are poorly understood. In 2014, New York City established a reporting system for bed bug infestation through the city's 311

Complaints (https://data.cityofnewyork.us/Housing-Development/Housing-Maintenance-Code-Complaints/uwyv-629c), Complaint Problems (https://data.cityofnewyork.us/Housing-Development/Complaint-Problems/a2nx-4u46), 311 Call Center Inquires (https://data.cityofnewyork.us/City-Government/311-Call-Center-Inquiry/tdd6-3ysr). Geographic Information for the study area is freely available for download at New York City Planning (https://www1.nyc.gov/site/planning/data-maps/open-data/census-download-metadata.page?tab=2). Census and community survey information used for this study are available for download at: The New York City Community Health Survey (https://www1.nyc.gov/site/doh/data/data-sets/community-health-survey.page) The American Community Survey, and The United States Census Bureau (https://www.census.gov/quickfacts/).

**Funding:** This work was supported by the National Socio-Environmental Synthesis Center under funding received from National Science Foundation Grant DBI-1639145. KPH was supported by the University of Pennsylvania Diversity Postdoc Fellowship and National Institute of General Medical Sciences of the National Institutes of Health under Award Number K12GM081259. The funders had no role in study design, data collection and analysis, decision to publish, or preparation of the manuscript.

**Competing interests:** The authors have declared that no competing interests exist.

database, a dedicated phone and online system to access NYC services and information [17, 18]. Additionally, in 2014, the New York City Department of Health and Human Services collected data on probable bed bug infestations, estimating a prevalence of 5.1% of households citywide, with some areas (defined by New York United Hospital Fund Regions) reporting up to 12% of households infested [19]. Other cities across the United States have experienced high levels of infestation and have responded with various policy and public health interventions [16, 20].

The public health burden of bed bug infestations is substantial. Not only are residents of infested dwellings subject to physical symptoms such as irritating and painful bites, rashes, sleep loss, and allergic reactions, but some also suffer immense psychological and emotional distress [6, 13, 21–23]. Residents of infested dwellings report increased anxiety and depression, linked both to the physicality of the infestation as well as the incurred social stigma [23, 24]. The health burden on the homebound and elderly is particularly relevant, as home healthcare personnel and social workers without adequate training can be reticent, or refuse to, enter infested areas [25, 26]. Additionally, due to the expense and difficulty of effective extermination [27], poisoning, property damage, and exposure to inexpertly applied insecticides has occurred [26, 28]. Bed bugs are competent hosts for *Trypanosoma cruzi* and *Bartonella quintana*, the etiologic agents of Chagas disease and trench fever respectively [6, 29, 30]. Whether bed bugs are, or could become, epidemiologically relevant in the transmission of these agents remains unclear.

To combat this public health crisis, New York City has instituted two bed bug disclosure policies. In 2010, New York passed its first ordinance that required landlords to report bed bug infestations occurring in the previous year to residents and prospective residents [31]. The city passed a second disclosure ordinance in 2017 requiring landlords to report annually all units infested or treated for bed bug infestation, and to notify all residents in the building, rather than only current or prospective tenets of a given unit [32].

Making use of data resulting from 311 and other reporting systems, we assess the spatial-temporal trends in bed bug complaints and inquiries made by New York City residents and building owners. In addition to exploring these trends, we question whether the policy-driven approaches to managing bed bug infestations have resulted in a decrease in the rate of complaints over time. We also question whether these policy approaches have had differential impacts across New York City's boroughs.

## Methods

### Databases

The area referred to as New York City is administratively organized into five "boroughs": Manhattan, Brooklyn, Queens, Bronx, and Staten Island (S1 Fig), which are each designated as their own counties by the State of New York. We report general characteristics and demographics for these boroughs in S1 Table. We examine three databases archived by New York City: bed bug inquiries registered by the city's 311 non-emergency reporting system, official bed bug complaints made to the city's Department of Housing Preservation and Development (HPD), and building owner reported bed bug infestations reported to the city's Department of Housing Preservation and Development. The specifics of each database (attributes, time scales, and geographic information) are summarized in S2 Table and described in detail below. All databases were analyzed individually, and no data sources were pooled during analysis.

## NYC Open Data and NYC 311: 311 inquiries and 311 bed bug specific requests

Since 2009, New York City's Open Data Portal has maintained an online database of information collected by the city government. One of the largest datasets is the 311 service, a designated system for non-emergency information and reporting, which allows individuals, organizations, and businesses to access New York City's government services and information [18]. We accessed all 311 data inquiries from 2010–2019 focusing on 311 inquiries specific to bed bugs (all bed bug related inquiries as well as official complaints).

## NYC housing maintenance code complaints: Official bed bug complaints

Under the Housing and Maintenance Code, tenants have the right to a bed bug-free environment [32]. Specifically, in the Housing and Maintenance Code, Subchapter 2, Article 4 names bed bugs as a Class B violation, meaning that the landlord is legally obligated to eradicate (*sic*) the problem within 30 days [33]. These official complaints were made publicly available in 2014 and are updated continuously. We compiled all Housing Maintenance Code Complaints from 2014–2019 [31] that were the result of a potential bed bug complaint, and, as a point of comparison, also compiled cockroach infestation complaints from the same data. Since New York City is currently the only city with mandatory reporting protocols operating at both the individual unit and building levels, and these laws were instated simultaneously and equally throughout the city, there is not an obvious area or other city that can be used as a standard 'control' to assess the impact of the bed bug laws. Cockroaches and flies, like bed bugs, are persistent urban pests whose control relies on communication between tenant and landlord. Importantly, cockroach and fly infestation reporting are identical to bed bug reporting. However, unlike bed bugs there is no new legislation or policies that specifically target cockroach or fly infestation. Based on these characteristics, we use cockroach and fly complaints as a control-type groups which will enable us to identify potential temporal changes in insect reporting, in the absence of a true control area or population.

## Building owner reports of bed bug infestation

The newest bed bug disclosure law requires property owners of multiple dwellings (buildings with 3+ residential units) to report annually the number of units infested with bed bugs or that were treated for bed bugs [32]. While this data is required to be accessible to the public, currently New York City does not have this data published through NYC Open Data. We obtained bed bug infestations reported by property owners for 2018 through a Freedom of Information Request. Data were reported at the building level and included information on the total number of residential units, the number that had experienced a bed bug infestation, the number of infestations treated, and the number of units re-infested.

## Statistical analyses

**Assessing the geographical distribution of bed bug infestation in New York City.** To assess the geographical distribution of bed bug complaints we calculated the number of bed bug complaints per Neighborhood Tabulation Area (NTA). NTAs are combinations of whole census-tract level population data with a minimum of 15,000 residents per aggregation [34]. The NTA codes for all NTA areas are displayed on S1 Fig with community names listed in S1 File. Unlike census tracts, which are prone to high sampling error, using NTAs as a geographic boundary helps to standardize areas by population while providing a more statistically reliable estimate of population [34]. We divided the total number of bed bug complaints by the

**Table 1. Modeling framework used to assess potential space-time interactions of bed bug complaints per NTA area.**

Model framework:

$\eta_{it} = b_0 + u_i + v_{i} + \gamma_t + \phi_t + \delta_{it}$

Where $b_0$ is the intercept which quantifies the outcome rate in the entire study region, $v_i$ the area-specific effect which is modeled as exchangeable, $u_i$ is the spatially-structured area-specific effect, $\gamma_t$ represents the temporally-structured effect, $\phi_t$ is a temporal unstructured effect specified as a Gaussian exchangeable prior, and lastly $\delta_{it}$ is a differential trend of the interaction between space and time.

| Model | Interaction | Parameters Interacting | Summary | DIC |
|---|---|---|---|---|
| Model 1 | I | $v_i$ and $\phi_t$ | Two unstructured effects interact. Assumes no spatial and/or temporal structure | 7404.7 |
| Model II | II | $v_i$ and $\gamma_t$ | Combines the temporally structured main effect and unstructured spatial effect. | 7500.7 |
| Model III | III | $\phi_t$ and $u_i$ | Combines the unstructured temporal effect and spatially structured main effect | 7480.4 |

estimated population for each year per 100,000. We mapped this information using QGIS software [35].

**Modeling the temporal patterns of housing maintenance code complaints for bed bugs and cockroaches.** We modeled the number of complaints per month as a harmonic function of time using linear regression with a linearly decreasing amplitude over time. Incorporating the harmonic function helped account for the high degree of seasonality observed. We standardized the total number of reported infestations to the total number official complaints made through 311. We repeated this analysis for each of the New York City boroughs, standardizing the number of complaints per borough per population size using population estimates obtained by the American Community Survey [17].

**Spatiotemporal model.** To assess the spatiotemporal dynamics of bed bug complaints, we first calculated the ratio between the observed and expected counts (Standard Incidence Ratio SIR) of bed bug complaints per NTA area [34]. The expected counts are the total number of cases expected if the population area (NTA area) behaved as the standard population behaves. Expected counts were calculated using indirect standardization, we computed the expected counts as:

$$E_i = r^{(s)} n^{(i)} \tag{1}$$

Where $r^{(s)}$ is the rate calculated as the number of cases divided by the total population in all areas, and $n^{(i)}$ is the population of area $i$ (the population of the NTA area). We then used a Bayesian Hierarchal modeling approach using Integrated Nested Laplace Approximation (INLA), that assessed the relative risk of bed bug complaints per NTA area, where the risk of the bed bug complaints in a given NTA area is compared to the expected number of complaints per NTA area given the population [36]. Relative risk is defined by the spatiotemporal model as the posterior mean of the spatial temporal interaction $\delta_{it}$ for bed bug complaints per NTA area. This approach enabled us to utilize information from neighboring NTA areas and incorporate space-time covariates as structured and unstructured random effects (S5 Table). Modeling framework is further discussed in Table 1. This spatial-temporal approach accounts for not only spatial structure using neighborhood dynamics, but also temporal and spatial-temporal interactions, and smooths or shrinks extreme values that would potentially result by using SIR values alone [37, 38].

**Comparing landlord reports of bed bug infestation to 311 resident complaints for 2018.** We examined the concordance between the 311-complaint data and recently available data on building owner reports for each NTA using linear regression.

## Results

### Summary of 311 inquiries linked to bed bugs from 2010–2019

From 2010–2019 there were a total of 72,701,278 inquiries processed by 311 either online, through the app, or by phone. Since 2010, the number of inquiries processed by 311 has

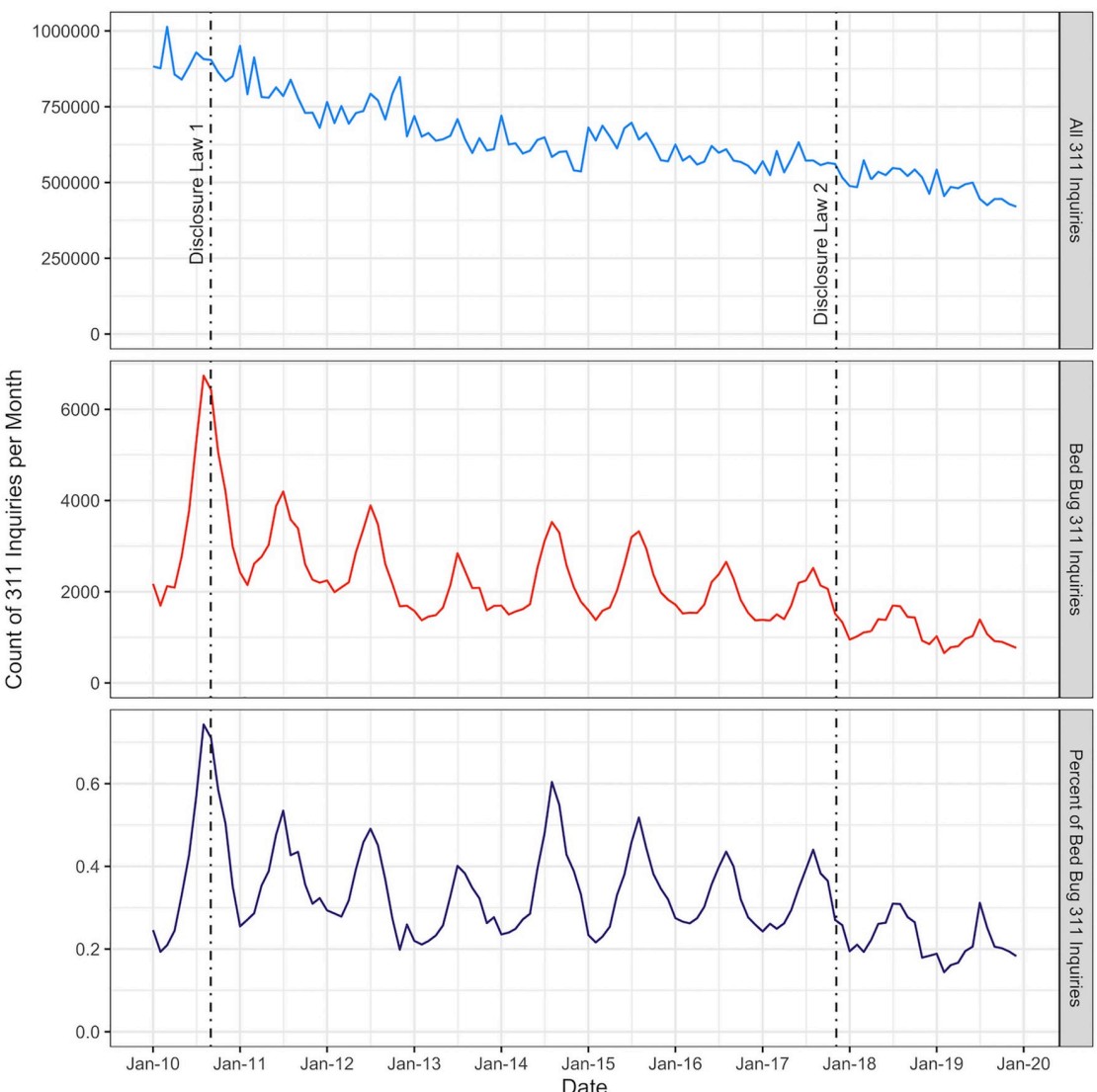

**Fig 1. New York City 311 data usage and bed bug related inquiries.** New York City 311 inquiries, bed bug related inquiries (which include official bed bug complaints registered to the Department of Housing Preservation, and bed bug related inquiries standardized by the total number of 311 inquiries, from 2010–2019. Bed bug related inquires were extracted using a text search algorithm. Dates of the two bed bug disclosure laws are indicated by a dashed line.

steadily decreased (Fig 1). We identified 185,289 inquiries specific to bed bugs. These inquiries were processed as 18 specific descriptions which were forwarded to seven different city departments or by the general 311 call center (S2 Table).

Like the general 311 inquiries, bed bug related inquiries decreased from 2010–2019. The largest peak in bed bug related inquiries occurred in August 2010 (n = 6,737). Throughout the time series, bed bug inquiries peaked during late summer (June–August) and decreased from September through April, creating a distinct seasonal pattern.

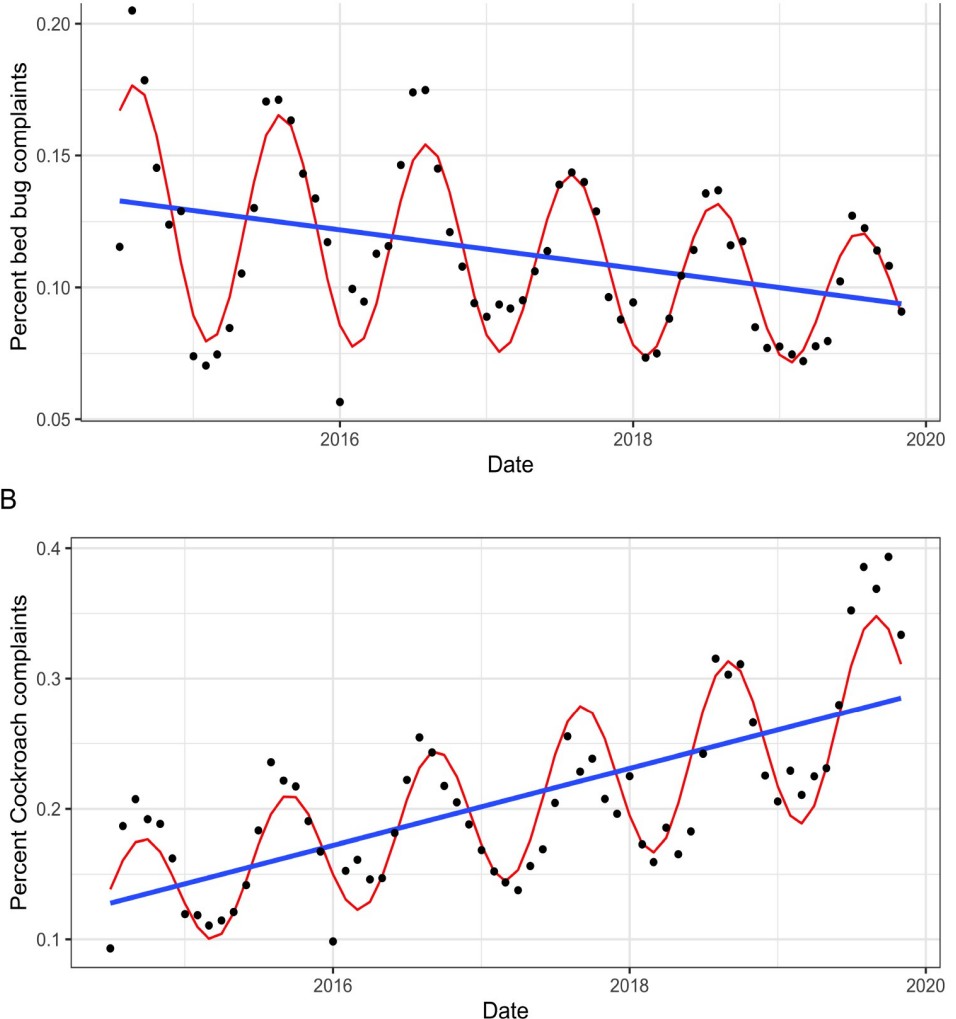

**Fig 2. Longitudinal analysis of official bed bug and cockroach complaints registered to HPD.** Graphical representation of the results of a linear harmonic model assessing the temporal relationship of official bed bug and cockroach complaints from 2014–2019. (*A*) Bed bug complaints modeled as a linear harmonic model with decreasing amplitude over time. (*B*) Cockroach complaints modeled as a linear harmonic model with increasing amplitude over time.

## Temporal patterns of housing maintenance code complaints for bed bugs, cockroaches, and flies

Official bed bug complaints followed regular seasonal patterns (Fig 2A). When this harmonic pattern was extracted, the residuals formed a linear decreasing trend from 2014–2019. The decreasing temporal trend was significant ($p < 0.001$), indicating that bed bug complaints significantly decreased from 2014–2019 (S3 Table). Cockroach complaints followed a similar seasonal pattern to bed bug complaints (Fig 2B). However, unlike bed bug complaints, we observed a significant positive temporal trend ($p < 0.001$) (S3 Table). Fly complaints however, remained constant throughout the study period only exhibiting seasonal increases like the patterns observed in bed bug and cockroach complaints (S2 Fig).

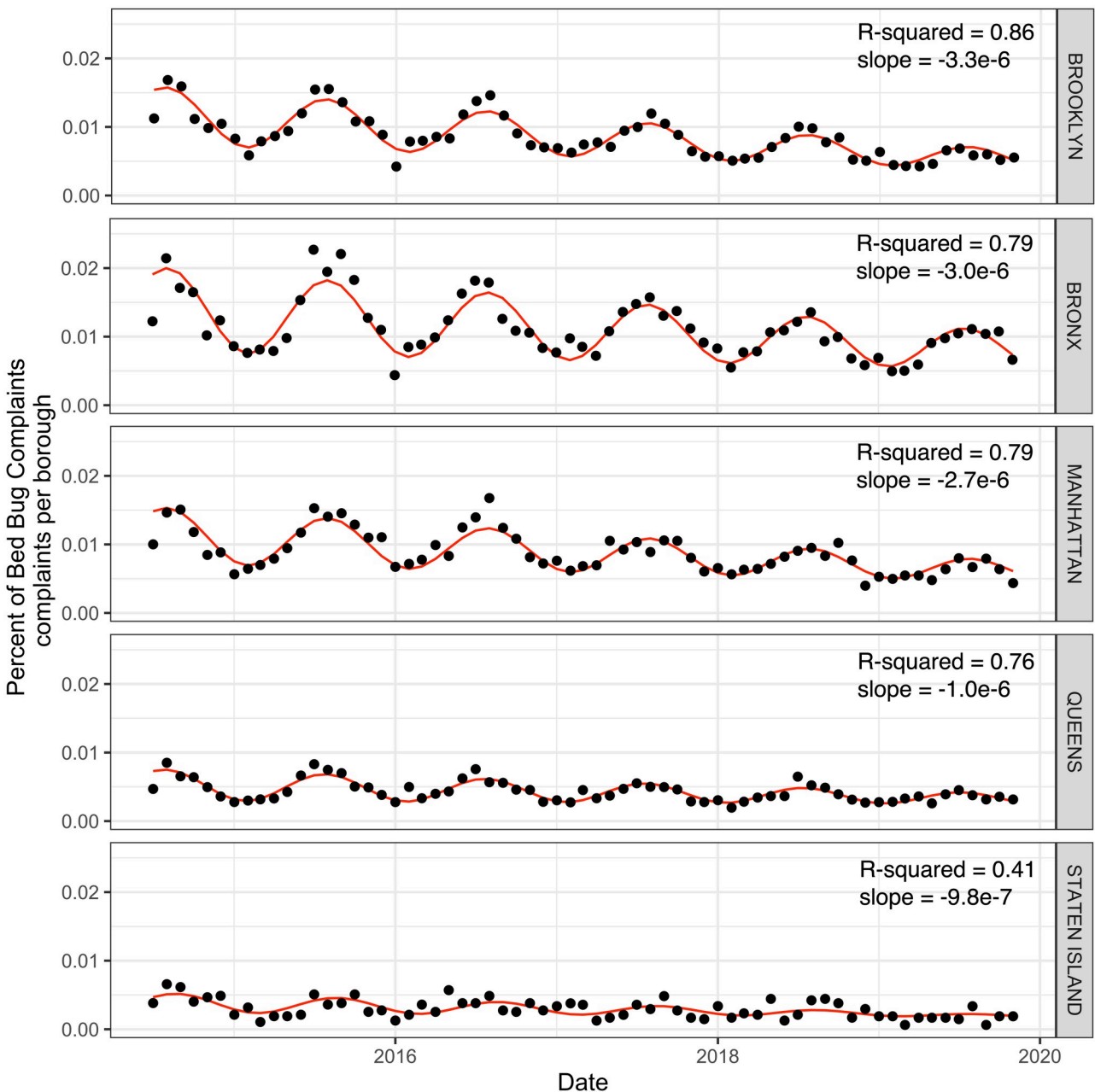

**Fig 3. Harmonic linear model assessing the temporal patterns of official bed bug complaints for each of the five NYC boroughs.** Graphical representation of model results assessing official bed bug complaints per each of the NYC borough standardized by borough population from 2014–2019. Model fit assessed by R-squared, and slope of the linear residual pattern are reported.

Across all the boroughs, bed bug complaints decreased (Fig 3). Brooklyn had the greatest yearly rate of decrease, followed by the Bronx and Manhattan, Queens and lastly Staten Island (S3 Table, S1 Fig). The seasonal pattern was evident across all the boroughs.

## Spatiotemporal modeling of bed bug complaints in New York City

From 2014–2019, the number of official bed bug complaints processed by HPD were widely distributed throughout the five boroughs (Fig 4). The Bronx had the greatest proportional

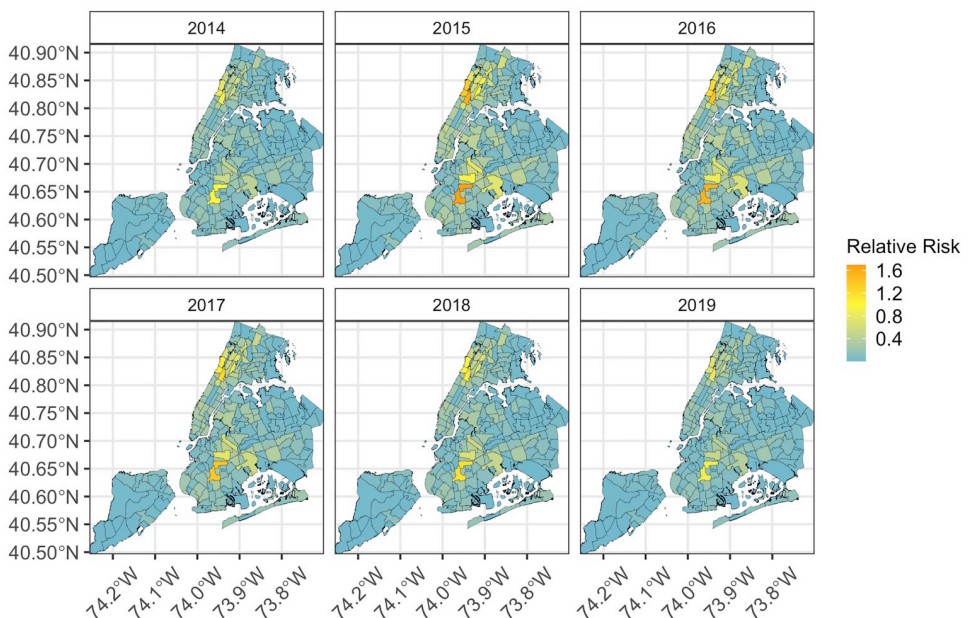

**Fig 4. Relative risk of official bed bug complaints per NTA area for New York City incorporating a temporally structured fixed effect.** This map demonstrates the results of the spatiotemporal model detailed in Table 2 and computes the relative risk of bed bug complaints per NTA area. Relative Risk is the posterior mean of the spatial temporal interaction $\delta_{it}$ for bed bug complaints per NTA area assuming no spatial and/or temporal structure. This spatiotemporal approach enables us to examine relative risk while adjusting for both space and time.

number of bed bug complaints followed by Brooklyn, Manhattan, Queens, and Staten Island for the study period (S4 Table). During 2015, bed bug complaints peaked and then decreased across all boroughs (S4 Table).

The non-parametric dynamic space-time model incorporated the addition of neighborhood effects and temporal effects where the relative risk of neighboring NTA areas influences the relative risk of each NTA area by year (Table 1). The relative risk in some NTA areas were influenced by temporal neighborhood effects. Of note, these effects were noticeable in Brooklyn in 2014, 2016, and 2017 (Fig 4 compared to S3 Fig), and were significant (Table 2), however these effects did not substantially change the overall patterns of Relative Risk when compared to the original Standard Incidence Ratio (SIR) estimates (S3 Fig). When space-time interactions were considered the Deviance Information Criterion DIC model fit did not improve, and the lowest DIC was observed (DIC = 7404.7) when these effects were not included

**Table 2. Results of the nonparametric spatiotemporal model (model 1) without space-time interaction effects.**

| Model I components | | Model hyperparameters | | | |
|---|---|---|---|---|---|
| **Fixed Effects** | **Definition** | **Mean Posterior** | | **0.025 quant** | **0.975 quant** |
| Intercept | Outcome rate in the entire study region | | -2.5 | -2.7 | -2.3 |
| Random Effects | | | | | |
| ID.area BYM model | spatial components | IID component | 0.4 | 0.4 | 0.5 |
| | | spatial component | 250.7 | 5.0 | 1520.0 |
| ID.year RW2 model | Temporally structured effect | | 19.0 | 3.8 | 58.0 |
| ID.year1 IID model | Temporally unstructured effect | | 24316.4 | 59.4 | 154000 |
| ID.area.year IID model | Area-year interaction index | | 13.7 | 11.3 | 160.0 |

(Table 1). Put differently, the space-time model (Model I Table 1) adequately captures the spatiotemporal trends and was not improved with interaction effects for space or time (Models II and III).

## Comparing landlord reports of bed bug infestation to 311 resident complaints for 2018

When we compared 2018 HPD bed bug complaint data (total number of complaints per NTA) to building owner reported bed bug infestation (number of reports per NTA) we found significant positive agreement (0.92 ± CI 0.81, 1.02, p < 0.001), suggesting that while the correlation was high ($R^2$ = 0.60), only a portion of the infestations may be officially reported.

## Discussion

The primary objective of this study was to characterize bed bug infestation in New York City and assess—in the absence of a true control population—if changes in bed bug infestation correspond temporally to changes in pest management policies employed by New York City. During the time periods assessed, both general 311 inquiries about bed bugs (2010–2019) and official bed bug complaints (2014–2019) decreased in New York city. This decrease was significant (p < 0.001) both when standardized by population and against the totality of 311 inquiries and their decline in reporting was not explained by insect reporting generally. When compared to patterns observed in cockroach and fly complaints, which stand as imperfect insect control populations, the decrease in bed bug complaints is notable. The contrasting pattern of bed bug complaints, as well as strong seasonal patterns, indicate that the decrease in complaints is not an artifact of overall 311 or reporting use as these trends continued to be significant following standardization. While surprisingly little is known regarding the seasonality of bed bugs [39], the seasonal trends observed where bed bug complaints peaked in the summer and decreased during the winter is consistent with other studies of bed bug seasonality [39–43] and other reporting trends noted in the United States and elsewhere [39, 44]. Ultimately, disclosure laws, new approaches in pest management [27], increased knowledge, and commitment to inspections have likely all contributed to this decrease in bed bug complaints in New York City.

New York City has enacted one of the most comprehensive strategies to combat bed bug infestations among major US cities [16]. Ordinances assign responsibility for treatment to landlords, and subsequent disclosure of infestations to tenants. Additionally, the city commits substantial resources to respond to all bed bug related complaints. While our study cannot individually and specifically assess the efficacy of these policies, the decrease in bed bug complaints across all boroughs provides evidence that they are working. A previous modeling study by Xie 2019 et al. [45], not only demonstrates that disclosure laws can reduce the spread of bed bug infestations, but also that they can reduce the costs incurred by landlords and tenants [45]. Strong disclosure laws, like those in New York City, may therefore offer a cost-effective road map for other cities struggling with bed bugs.

Best practices for bed bug management have also improved over the course of the bed bug epidemic [27] and could account for the decrease in complaints. These improvements are not specific to New York. There are no comparable studies from other major cities—if such studies were to show a similar downward trend it might be reasonable to attribute the decrease in 311 complaints in New York to improved pest management methods alone. However, without sufficiently large samples of other cities with the same types of disclosure laws, it is difficult to assess if specific bed bug management practices or legislation account for the decreases seen in New York City. Governmental and nongovernmental entities have also increased educational efforts, and these may have also improved knowledge among landlords and residents

(although this has not been formally assessed). Residents may choose to work directly with their landlord and not involve the city. If this were the case, we might expect to see areas in which mandated landlord reporting was out-of-step with 311 inquiries. Instead, we saw high concordance between the two ($R^2$ = 0.60).

The decrease in bed bug complaints was observed across all New York City boroughs. However, the rate of decline was not equivalent. Higher income boroughs (Manhattan and Brooklyn) saw steeper declines than the lower income boroughs of Queens and Staten Island. The differences in the rates of decline are very likely due to differences in financial means to properly treat infestations and incorporate recommendations of Integrated Pest Management (IPM) [27, 45]. Differences in trust and access to city government and services are also likely to affect rates of decline [16, 20, 46–49]. In Chicago, bed bug infestation was highly associated with lower-income neighborhoods, crowding, and eviction notices. Additionally, inexpertly applied IPM new and reintroduction of bed bugs, and high rates of pyrethroid resistance may result in chronically infested areas particularly in lower-income areas [9]. While we did not specifically assess sociodemographic features, it is likely that similar patterns exist in New York City and indeed have been noted in smaller scale surveys and assessments [47, 48].

Despite substantial decreases in all boroughs, on the finer scale of NTD areas there are many persistently infested areas (Fig 3). Many, though not all, of these persistently infested areas are in lower-income areas. While New York City has made a vested effort to emphasize that tenants are not financially responsible for bed bug treatment, fear of eviction or cost (which are substantial) may prevent tenants from reporting bed bug infestation, which may promote spread to non-infested units. Areas with limited financial resources are therefore potentially at risk for persistent or entrenched bed bug infestation.

Our study was not without limitations. Self-reported and landlord reporting of bed bug infestation have inherent biases and inaccuracies [47]. It is possible that tenants have started to report fewer complaints through the city and instead communicate directly with their building manager. However, for the records we have available we see high concordance between infestations reported by building managers and those reported to the city by tenants. Sociodemographic differences have been documented between bed bug complaints and confirmed bed bug violations, with non-verified bed bug complaints (complaints that resulted in a negative bed bug inspection) occurring primarily in higher-income, majority white non-Hispanic neighborhoods [47]. Despite not incorporating sociodemographic information into this modeling framework, we still captured substantial variation in bed bug complaints across NTA areas. However, barring a standardized spatial sampling design and comparable control populations, the data provided by New York City Open Data is likely the largest and most complete proxy available to estimate the spatial and longitudinal patterns in bed bug infestations.

Bed bugs are tied inextricably to their human hosts and the dynamic urban environment. Their resurgence in urban spaces and the necessity of rapid intervention strategies have elicited health policy, increased monitoring, and novel treatment strategies over the past 10 years. While there is continued need to increase active surveillance for bed bug infestations, particularly among vulnerable populations, the policy and public health approaches employed by New York City appear to be a step in the right direction.

## Supporting information

**S1 Fig. New York City boroughs and NTA area map.** A map displaying the NTA areas of New York City within their respective boroughs. NTA areas are designated with their NTA code, the respective neighborhoods associated with each NTA code are available in S1 File. (TIF)

**S2 Fig. Longitudinal analysis of official fly registered to HPD.** Graphical representation of the results of a linear harmonic model assessing the temporal relationship of official fly complaints from 2014–2019. Fly complaints modeled as a linear harmonic model with decreasing amplitude over time.
(TIF)

**S3 Fig. Standard Incident Ratio (SIR) of official bed bug complaints per NTA area for all New York City boroughs from 2014–2019.** Standard Incident Ratio was calculated as the ratio between the observed and expected number of bed bug complaints per NTA area. Expected accounts were calculated via indirect standardization. Specific NTA area names per borough are listed in S1 File.
(TIF)

**S4 Fig. Plotted posterior mean of the BYM random effect of the from 2014–2019.**
(TIF)

**S5 Fig. Comparison between building manager reported bed bug infestation and official bed bug complaints registered to HPD for 2018.** Correlation between official bed bug complaints from residents (n = 6376) and building manager reported infestation (n = 7303) was high ($R^2 = 0.60$).
(TIF)

**S1 Table. New York City boroughs and basic statistics.** Basic demographic statistics for the five boroughs of New York City. Information for table was obtained through the U.S. Census Bureau QuickFacts resource [49].
(DOCX)

**S2 Table. Descriptive properties of each of the datasets.** Database descriptions including timeframe, georeferencing information, organizational management of database, and codes used for analysis.
(DOCX)

**S3 Table. Total number of inquiries processed by New York City's 311 that included bed bugs as part of the description from 2010–2019.** Description of bed bug related inquires and the agencies that processed the request.
(DOCX)

**S4 Table. Association between time and official bed bug and cockroach complaints accounting for seasonality.** Model results of a linear harmonic model assessing the association between month and number of official bed bug complaints and cockroach complaints from 2014–2019. Bed bug and cockroach complaints were standardized by the total number of 311 inquiries to obtain percentages.
(DOCX)

**S5 Table. Bed bug complaints throughout the boroughs as processed by HPD by year.**
(DOCX)

**S1 File. New York City NTA area codes and their respective names.** Datasheet that shows summary statistics for each NTA mapped in S1 Fig and denotes the district name for each NTA area.
(CSV)

## Acknowledgments

We thank the other participants in the workshop "Socio-Spatial Ecology of the Bed Bug and its Control"—Claudia Arevalo, Dawn Biehler, Stephen Billings, Warren Booth, Ludovica Gazze, Loren Henderson, Sara McLafferty, Shannon Sked, and Chris Sutherland—for discussions that helped shape this paper.

## Author Contributions

**Conceptualization:** Kathryn P. Hacker, Andrew J. Greenlee, Alison L. Hill, Daniel Schneider, Michael Z. Levy.

**Data curation:** Kathryn P. Hacker, Daniel Schneider, Michael Z. Levy.

**Formal analysis:** Kathryn P. Hacker, Alison L. Hill.

**Funding acquisition:** Michael Z. Levy.

**Investigation:** Kathryn P. Hacker, Alison L. Hill, Daniel Schneider, Michael Z. Levy.

**Methodology:** Kathryn P. Hacker, Andrew J. Greenlee, Michael Z. Levy.

**Project administration:** Daniel Schneider, Michael Z. Levy.

**Resources:** Andrew J. Greenlee, Alison L. Hill, Daniel Schneider, Michael Z. Levy.

**Supervision:** Andrew J. Greenlee, Daniel Schneider, Michael Z. Levy.

**Validation:** Andrew J. Greenlee, Alison L. Hill, Michael Z. Levy.

**Visualization:** Kathryn P. Hacker.

**Writing – original draft:** Kathryn P. Hacker.

**Writing – review & editing:** Kathryn P. Hacker, Andrew J. Greenlee, Alison L. Hill, Daniel Schneider, Michael Z. Levy.

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
