## [Decision Letter · Decision Letter 0]

16 Dec 2021

PONE-D-21-35090Spatiotemporal Trends in Bed Bug Metrics: New York CityPLOS ONE

Dear Dr. Hacker,

Thank you for submitting your manuscript to PLOS ONE. After careful consideration, we feel that it has merit but does not fully meet PLOS ONE’s publication criteria as it currently stands. Therefore, we invite you to submit a revised version of the manuscript that addresses the points raised during the review process.

ACADEMIC EDITOR: Correct the references according to the recommended format of the JournalAdd latest references as pointed out in the reviewer’s commentsCheck the supplementary data according to the reviewer's commentsImprove the conclusive remarks

We look forward to receiving your revised manuscript.

Kind regards,

Javaid Iqbal, PhD

Academic Editor

PLOS ONE

Journal Requirements:

[We thank the other participants in the workshop “Socio-Spatial Ecology of the Bed Bug and its Control”—Claudia Arevalo, Dawn Biehler, Stephen Billings, Warren Booth, Ludovica Gazze, Loren Henderson, Sara McLafferty, Shannon Sked, and Chris Sutherland—for discussions that helped shape this paper. This work was supported by the National Socio-Environmental Synthesis Center under funding received from National Science Foundation Grant DBI-1639145. KPH was supported by the University of Pennsylvania Diversity Postdoc Fellowship and National Institute of General Medical Sciences of the National Institutes of Health under Award Number K12GM081259.]

 [This work was supported by the National Socio-Environmental Synthesis Center under funding received from National Science Foundation Grant DBI-1639145. KPH was supported by the University of Pennsylvania Diversity Postdoc Fellowship and National Institute of General Medical Sciences of the National Institutes of Health under Award Number K12GM081259.]

Reviewers' comments:

Reviewer's Responses to Questions

**Comments to the Author**

1. Is the manuscript technically sound, and do the data support the conclusions?

Reviewer #1: Yes

Reviewer #2: Yes

Reviewer #3: Yes

2. Has the statistical analysis been performed appropriately and rigorously? 

Reviewer #1: Yes

Reviewer #2: Yes

Reviewer #3: Yes

3. Have the authors made all data underlying the findings in their manuscript fully available?

Reviewer #1: Yes

Reviewer #2: Yes

Reviewer #3: Yes

4. Is the manuscript presented in an intelligible fashion and written in standard English?

Reviewer #1: Yes

Reviewer #2: Yes

Reviewer #3: Yes

5. Review Comments to the Author

Reviewer #1: The New York City area has thousands of licensed pest management professionals and hundreds of pest control companies. Could these results been explained by a mere shift from the public to the private sector?

Minor Comments

Reference 3 looks incomplete (year of publication).

References 1-4: more recent references could be added.

Could one single bedbug complaint be registered in more than one database?

Why was the year 2020 not included? It could have been interesting to study the effect of confinement.

Reviewer #2: Dear Authors

After going through the MS, "Spatiotemporal Trends in Bed Bug Metrics: New York City", I can conclude that it contains very useful information for readers especially the involved in Vectors of Disease Control. Research explained in the MS is well designed and the well constructed. I don't see any problem with data presentation and the description of results and discussion.

In suppl. material file, Suppl. Table 5 is repeated with titles "Modeling framework used to assess potential space-time interactions of bed bug complaints per NTA area" and "Results of the nonparametric spatiotemporal model (model 1) without space-time interaction effects". These might be Tables 5 and 6. I suggest that Tables 5 and 6 may be included in the results of main MS along with Suppl. Figs 1 and 3. However, it depends if the authors agree.

Reviewer #3: Overall, the authors’ approach to analyze the bed bug trend in NYC is appropriate. My only concern is about the main conclusion: the number of bed bugs complains decreased, while cockroach complaints increased from 2014 based on data period of 2010-2019. The decrease in bed bug complaints is not surprising. Increased awareness, better methods and materials and control strategies, regulations, fatigue in reporting bed bugs might have contributed to the decline of bed bug complaints. The conclusion about increasing cockroach complaints is surprising. There is no evidence from pest control industry that shows cockroach infestations are increasing in the last 5-7 years. There are no revolutionary novel products appeared for cockroach control since 2014. I do not see a reason why cockroach infestations increased after 2014. Authors need to re-investigate the data and interpret the results more cautiously.

Other minor comments:

Line 90-96. Need to describe the 5 boroughs being studied.

Line 165. What does the relative risk scale mean? Need to explain.

Line 192. Supplemental table 4 does not show seasonal data.

Line 200. Supplemental Fig. 1 does not show the names of these boroughs.

Line 227. "R2" The number 2 should be in upper case

Line 245-246. Delete the discussion about canine. It is not an effective bed bug detection method.

References. Need to check the styles.

Fig. 4 legend. Need to be more descriptive. Please indicate this is for NYC. Explain the scales. This figure has lower resolution than the supplementary Figure.

Supplemental Table 2. Need to add more details so readers know this is about NYC and only during 2020-2019. This Table shows 46886 bed bug complaints from residents to HPD during 2020-2019. If divided by 10, the number is still very large. Why supplemental figure shows very low number of complaints in 2018?

Supplemental Table 3. “Cockroach complaints” need to be moved further to the left.

Supplemental Fig. 1. Use smaller font for x-axis legend. The “standard incidence ratio” legend needs to match with that for Fig. 4.

6. PLOS authors have the option to publish the peer review history of their article (what does this mean?). If published, this will include your full peer review and any attached files.

Reviewer #1: No

Reviewer #2: **Yes: **Rashad Rasool Khan

Reviewer #3: No

---

## [Author Response · Author response to Decision Letter 0]

25 Feb 2022

We thank the reviewers and editorial staff for their comments. We have addressed each of the concerns in the Responses to Reviewers document.

---

## [Decision Letter · Decision Letter 1]

7 Apr 2022

PONE-D-21-35090R1Spatiotemporal Trends in Bed Bug Metrics: New York CityPLOS ONE

Dear Dr. Hacker,

Thank you for submitting your manuscript to PLOS ONE. After careful consideration, we feel that it has merit but does not fully meet PLOS ONE’s publication criteria as it currently stands. Therefore, we invite you to submit a revised version of the manuscript that addresses the points raised during the review process.

We look forward to receiving your revised manuscript.

Kind regards,

Javaid Iqbal, PhD

Academic Editor

PLOS ONE

Journal Requirements:

Additional Editor Comments:

Minor revision (See reviewer's comments)

**Reviewers' comments:**

**Comments to the Author**

Line 273. Add a space before =.

Reference section: The scientific names (Cimex lectularius) need to be in italic font.

Line 426-428. Two references with same number. One of them should be 7

Line 525. Missing page number.

Line 538. May need to delete "Jul-Aug". Also delete "(1)".

---

## [Author Response · Author response to Decision Letter 1]

29 Apr 2022

Academic Editor:

Journal Requirements:

Please review your reference list to ensure that it is complete and correct. 

We have assessed the reference list and corrected the numbering which contained non-sequential numbering, and corrected typographical errors found in the references. 

Additionally, we added two references which were more updated than the original listed references. The articles added are: 

Bedbugs: Information for Tenants and Building Owners - NYC Health. [cited 25 Apr 2022]. Available: https://www1.nyc.gov/site/doh/health/health-topics/bedbugs-information-for-landlords-and-building-managers.page

Doggett SL, Geary MJ, Russell RC. The Resurgence of Bed Bugs in Australia: with Notes on Their Ecology and Control. Environmental Health 2004;4: 30-38.

If you have cited papers that have been retracted, please include the rationale for doing so in the manuscript text or remove these references and replace them with relevant current references. Any changes to the reference list should be mentioned in the rebuttal letter that accompanies your revised manuscript. If you need to cite a retracted article, indicate the article’s retracted status in the References list and also include a citation and full reference for the retraction notice.

While we did not note any journal retractions in our citation list, the article below did have a posted correction which updated their funding information. We added a “corrected version” to our citation list for clarity. 

Leulmi H, Bitam I, Berenger JM, Lepidi H, Rolain JM, Almeras L, et al. Competence of Cimex lectularius Bed Bugs for the Transmission of Bartonella quintana, the Agent of Trench Fever. Corrected Version. PLoS Negl Trop Dis. 2015;9: e0003789. doi:10.1371/journal.pntd.0003789

Reviewer Comments: 

We corrected all the following minor revisions noted by the authors. 

 Line 273. Add a space before =.

Corrected 

Reference section: The scientific names (Cimex lectularius) need to be in italic font.

Corrected 

Line 426-428. Two references with same number. One of them should be 7

Corrected, and fixed additional numbering errors in the citation list. 

Line 525. Missing page number.

Corrected 

Line 538. May need to delete "Jul-Aug". Also delete "(1)".

Corrected

---

## [Editor Report · Decision Letter 2]

9 May 2022

Spatiotemporal Trends in Bed Bug Metrics: New York City

PONE-D-21-35090R2

Dear Dr. Hacker,

We’re pleased to inform you that your manuscript has been judged scientifically suitable for publication and will be formally accepted for publication once it meets all outstanding technical requirements.

Kind regards,

Javaid Iqbal, PhD

Academic Editor

PLOS ONE
---

## [Editor Report · Acceptance letter]

16 May 2022

PONE-D-21-35090R2 

Spatiotemporal Trends in Bed Bug Metrics: New York City 

Dear Dr. Hacker:

I'm pleased to inform you that your manuscript has been deemed suitable for publication in PLOS ONE. Congratulations! Your manuscript is now with our production department. 

Kind regards, 

on behalf of

Dr. Javaid Iqbal 

Academic Editor

PLOS ONE